# Structural Features of the Fragments from Cast Iron Cauldrons of the Medieval Golden Horde: Neutron Tomography Data

**DOI:** 10.3390/jimaging9050097

**Published:** 2023-05-11

**Authors:** Bulat Bakirov, Veronica Smirnova, Sergey Kichanov, Eugenia Shaykhutdinova, Mikhail Murashev, Denis Kozlenko, Ayrat Sitdikov

**Affiliations:** 1Frank Laboratory of Neutron Physics, Joint Institute for Nuclear Research, 141980 Dubna, Russia; bulatbakirov4795@gmail.com (B.B.); veronicasm@jinr.ru (V.S.); denk@nf.jinr.ru (D.K.); 2Institute of Physics, Kazan Federal University, 420008 Kazan, Russia; eugeh@mail.ru (E.S.); ayrat.sitdikov@kpfu.ru (A.S.); 3Institute of Archeology Named after A. Kh. Khalikov, Tatarstan Academy of Sciences, 420012 Kazan, Russia; 4Institute of Aviation, Land Transportation and Power Engineering, Kazan National Research Technical University Named after A. N.Tupolev, 420111 Kazan, Russia; 5National Research Center “Kurchatov Institute”, 123098 Moscow, Russia; mihail.mmm@inbox.ru

**Keywords:** neutron tomography, porosity, cultural heritage, cast iron materials, medieval Golden Horde

## Abstract

The spatial arrangement of the internal pores inside several fragments of ancient cast iron cauldrons related to the medieval Golden Horde period was studied using the neutron tomography method. The high neutron penetration into a cast iron material provides sufficient data for detailed analysis of the three-dimensional imaging data. The size, elongation, and orientation distributions of the observed internal pores were obtained. As discussed, the imaging and quantitative analytical data are considered structural markers for the location of cast iron foundries, as well as a feature of the medieval casting process.

## 1. Introduction

Changes in the historical and cultural way of life of national groups and communities can be clearly reconstructed by the evolution of processing technologies, such as, in particular, the casting and forging of metal products [1]: utensils, jewelry, weapons, etc.

Metal casting fragments are excellent markers for reconstructing the features of technological levels of medieval settlements [2,3,4,5,6,7]. Now, there has been a growing interest in the research of archaeological and cultural materials using natural scientific methods [1,6,7,8,9]. Among them, the methods for non-destructive structural diagnostics should be noted, which are very careful with valuable objects of cultural heritage [10,11,12,13,14,15]. Currently, structural studies on archeological metal materials are focused on the search for some structural markers, which can help restore information about the ore mine or blacksmith locations, ways of importing metal raw materials, and identifying features of casting processes [16,17,18]. From this viewpoint, cast iron products are not only cultural and historical items but also convenient model objects for studying the processes of corrosion propagation and cracks in metal products, features of casting and stamping, and methods for the chemical processing of metal products.

Conventional metallography, X-ray fluorescence analysis, and scanning electron microscopy methods provide detailed structural data on elemental and phase composition but only from the surface or mechanical cut section of the iron-casting object [16,19,20,21,22]. This is due to the superficial thickness (up to several micrometers) of the penetration into the volume of the metal product. From this perspective, the neutron structural methods for non-destructive diagnostics, such as neutron tomography and neutron diffraction, provide a sufficient penetration depth into massive metal objects [13,14]. Therefore, research on the structural features of ancient cult products [12,23], jewelry [24,25,26], weapons [27,28,29], and coins [30,31,32] was performed successfully. Neutron imaging and neutron diffraction have made it possible to separate martensitic and austenitic phase components in steel weapons [33], to detect fasteners inside the handle of medieval swords [11] and defects and cracks in metal writing utensils [27], and to complete the non-destructive structural testing of a spearhead from a West Hungarian archaeological site [34]. However, the structural markers or benchmarks on the micron-scale level for large metal fragments and objects are debated [35,36]. As an example, large cavities or pores were found inside an ancient bronze axe [35]. These pores were associated with accompanying gas formation during the temperature treatment of arsenic–copper ore. The appearance of hydrogen or nitrogen gas can be associated with a prolonged contact of liquid metal with air, the presence of moisture in the iron casting mold, and the additives introduced into the metal forms [37,38,39,40]. As another example, the structural markers of technological processes in ancient pottery workshops were studied using X-ray and neutron tomography methods [41]. The orientation effects of inner pores and voids in the ancient ceramics were measured, and a comparative analysis of the simple fabrics was performed.

In our work, we tried to identify the structural features of fragments from cast iron cauldrons of the medieval Golden Horde [3,4,42,43], as a representative of the products from the ancient iron casting process. We have selected two groups of iron-cast fragments from the Selitrennoye settlement in the lower parts of the Volga River and the Bolgar settlement in the central Volga region. During the early Middle Ages, one of the first early feudal states in Eastern Europe, Volga Bulgaria, was formed on the middle Volga and in the lower reaches of the Kama River. In the VIII century, ancient tribes moved here from the lower reaches of the Volga, from the Azov steppes. Since the tenth century, feudal relations have been developing in Volga Bulgaria; castles with dependent rural populations in the district have been built; and ties with neighboring states, including Russian principalities, have been established [4,42]. Volga Bulgaria was one of the states that controlled trade on the Volga. The northern branch of the Silk Road passed through it. Since the tenth century, political and economic ties have been established on the land of Volga Bulgaria along the Upper Volga route with Russia, with the countries of the Caucasus and Central Asia, and through them with the countries of the Middle East. Bolgar became the first capital of the Golden Horde from the middle of the XIII century to the first half of the XIV century. During the heyday of Bulgaria, it was distinguished by a high level of landscaping. Archaeologists have identified the urban hydraulic structures, consisting of catchment basins, wells, and canals.

The Selitrennoye settlement is a leftover old Saray-Batu city. The city was founded by Batu Khan in 1254. At the beginning of the XIV century, it was another capital of the Golden Horde. There is a great city with continuous rows of houses, large mosques, palaces whose walls sparkle with mosaic patterns, reservoirs filled with clear water, extensive markets, and warehouses. The Khan’s palace stood on the highest hill above the bank of the Akhtuba River. Now, in the area of the Selitrennoye settlement, tiles with bright oriental ornaments, coins from the XIII to XIV centuries, fragments of ceramics, and clay water pipes were found. The city had its own pottery, foundry, jewelry, and blacksmith workshops [43]. 

Both settlements were capitals of the Golden Horde during the 13–15th centuries AD, but the difference in geographic location and historical area appoints them as manufactory centers with modified technological approaches for casting and processing cast iron within the borders of one ancient state in a similar historical period. We expect [4] a difference in the structural features of the internal structure of the fragments from cast iron cauldrons. The size, morphological, and orientation distributions of the inner structural elements in the thickness of the studied fragments were obtained using the neutron tomography method.

## 2. Materials and Methods

The metal samples under study are ten fragments from cast iron cauldrons, which date back to the 13–15th centuries AD. Photos of fragments N 25, 48, 65, 69, and 73 from the Bolgar settlement are presented in Figure 1. For extended labeling of the samples, we applied a mixed label with the official collection number of the sample and the coded location of the archaeological work: the Bolgar settlement (BS) and the Selitrennoye settlement (SS). Photos of fragments N 3, 5, 6, 7, and 8 from the Selitrennoye settlement are shown in Figure 2. These cast iron samples are larger than ones from the Bolgar settlement. The average length is in the range of 3–10 cm, and the thickness is up to 10 mm. The metal surface of all fragments is covered with corrosion.

The method of neutron tomography was used to study the structure and inner volume of the cast iron fragments. The fundamental difference in nature of neutron interaction with matter compared to X-rays provides additional benefits to neutron methods, including sensitivity to light elements, a notable difference in contrast between isotopes, and a high penetration effect through metals or heavy elements [12,14]. High neutron penetration into an iron material can provide several structural and morphological features from the analysis of reconstructed 3D models based on experimental tomography data [14]. The neutron imaging experiments were performed at the DRAGON facility for neutron radiography and tomography [14,44] using the IR-8 research reactor at the National Research Center “Kurchatov Institute” (Moscow, Russia). The *L*/*D* parameter of the facility collimator was 150. The double monochromator with pyrolytic graphite crystals in 002 reflection forms the neutron beam with a wavelength in the range of 1.75–2.00 Å. Pyrolytic graphite crystals, with a mosaicity of about 2°, are applied in the double-crystal monochromator. In our neutron imaging experiments, the neutron wavelength was λ = 2.4 Å. The neutron flux in the sample position was Φ ~ 3.6 × 10^6^ n/(cm^2^ s). The beam size on the sample was 7 × 7 cm. The spatial resolution of the obtained images was about 200 µm with a pixel size of 65 × 65 µm. The sample table is a load-bearing module for high-precision relative placement of the sample and detector along the neutron beam direction. The detector with removable scintillator plates based on a CCD camera ATIK-4000 with a Nikon lens (Nikon (Russia) LLC, Moscow, Russian Federation) was used for neutron image collection. For the reconstruction of three-dimensional (3D) models, sets of 360 angular image projections with a step of 0.5 degrees were collected. The typical exposure time for obtaining one neutron image was 60 s.

Three-dimensional data were reconstructed using the Simultaneous Iterative Reconstruction Technique (SIRT) algorithm [45] in the SYRMEP Tomo Project software package (version 1.6.3., https://github.com/ElettraSciComp/STP-Gui/releases/ (accessed on 7 May 2023)) [46]. Iterative reconstruction algorithms for tomography have demonstrated promising results in the ability to compute high-quality 3D images from less data. In this case, the application of iterative algorithms for tomography reconstruction calculations allows qualitative data to be obtained. We ran 150 iterations of the SIRT algorithms for neutron imaging data. A reduction in the calculation time was possible by using professional graphics cards with CUDA technology support.

For 3D neutron tomography data used in studies of porous materials, the choice of pore segmentation procedures is extremely important [47]. There are a large number of segmentation methods and their modifications used for image analysis in various fields of research. The quality of pore segmentation in 3D reconstructed data suffers significantly due to the effects of blurring and artifacts in the neutron tomography data. The corrected image must be binarized using a global threshold. The effect of uneven lighting or other artifacts on neutron imaging data does not allow us to reliably select a global threshold. The desired boundaries between metal and pores can be calculated using watersheds (WSs). The intensity of the image along the contours of the watershed varies and does not correspond to a single gray level. Therefore, the presence of false local maxima in the gradient image due to noise and blurring can lead to false segmentation. The effect of uneven lighting can be reduced by increasing the overall contrast of the image by increasing the gradients at the phase boundaries. The purpose of pore segmentation is to reliably determine the boundaries between phases. In grayscale images, such a border can be selected as the location of an array of points with the largest gradient. The problem of segmentation of phase boundaries using a gradient image is effectively solved using the WS transformation. In our study, we propose to introduce a marker image for pore segmentation. Such an algorithm is the addition of a binary image showing the approximate location of the pores on the original image. This can be represented as an image of seeds located at the pore locations. Based on the fact that each pore can be segmented by its own threshold, we propose to use a modification of the traditional approach to the watershed using a gradient image. The idea of the method is to calculate a local threshold for each of the pores that have been segmented with a watershed. The threshold value is defined as the minimum among the gray values of the original or filtered image belonging to the corresponding watershed lines. This operation helps to prevent, or at least minimize, the false segmentation that occurs due to the presence of incorrect maxima in the gradient image. In our neutron image studies, we added additional steps to the conventional WS method. This involves marking the pores segmented with the usual WS for each labeled pore and calculating the threshold as the minimum value of the original image on the corresponding watershed line or simply on the boundary of the pore, which is a binarization of the original image. This procedure was described in more detail earlier [47].

The visual representation and analysis of reconstructed data were performed using the VGStudio MAX 2.2 software (Volume Graphics, Heidelberg, Germany).

## 3. Results

In addition to the segmentation of the cast iron components and the surface layer of corrosion on the studied fragments (Figure 3a and Appendix A), it was possible to unambiguously distinguish certain structural features from the neutron tomography data. This is a developed pore space (Figure 4a and Appendix A). Moreover, if in most cast iron fragments from the Bolgar settlement, a uniform distribution of material inside the volume of the samples (Figure 3b) was observed with some single large pores or voids, then in almost all fragments from the Selitrennoye settlement, a large number of closed pores were found (as an example, Figure 4). However, it should be noted that many pores were also observed in the BS-65 sample from the Bolgar settlement. In addition, an oblong linear area with neutron attenuation coefficients other than those of cast iron material was observed in the sample BS-25 (Figure 3a). It is assumed that this local area of the cauldron was burned with fire or other high-temperature influence [4,48,49]. This is indicated by the flat pieces of the peeled material. The difference in neutron attenuation coefficients is associated with strong oxidation processes in this area of the cauldron.

The volumetric and morphological features of the pores inside the cast iron fragments were analyzed using the obtained 3D neutron data. The local threshold from the watershed approach [47] was used for the pore segmentation. The volumes of the reconstructed 3D models, the volumes of pores, and the calculated porosities for the cast iron fragments are listed in Table 1.

In the first view, a high porosity is observed only in the fragments from the Selitrennoye settlement; simultaneously, the samples from the Bulgar area have almost no internal large pores, except for the BS-65 sample. As an intermediate conclusion, the presence or absence of pores inside the volume of the archaeological cast iron materials or products can be distinguished as a structural marker. The algorithms and calculations for analyzing the 3D models of the fragments make it possible to obtain some morphological parameters of separated pores, as well as their mutual spatial orientation. The sizes of pores were estimated using the equivalent diameter parameter, which corresponds to the diameter of a sphere with the same volume as a pore [47]. The obtained distributions of the equivalent diameters of the pores for samples SS-3, SS-5, SS-6, SS-7, SS-8, and BS-65 are presented in Appendix A. Most of the pore sizes fall into the range of 0.8–2 mm, but several large pores are in the range of 5–8 mm. It should be noted that the presence of a great number of small pores in the cast iron fragment BS-65 from Bulgar was observed. It is assumed that the corresponding cauldron could have been brought through the river trade route [4,49], which was actively developing at this time. However, the statistical sampling is very low, so it is redundant to draw more accurate conclusions. For a more detailed analysis of the pore size distributions, we used the approximation of experimental data using the probability density function with a non-parametric estimation method for kernel density estimation with Silverman bandwidth [50,51,52]. In the frame of this approach, we can compare the calculated distributions from sample to sample. The calculated approximating curves are shown in Figure 5. It can be seen that the narrowest distribution is characteristic of the sample SS-5. Interestingly, for other samples, there is an expansion in the approximation curves with a corresponding shift into the large size range of the average equivalent diameter of the segmented pores (Figure 5). If the broadening in the size distribution indicates an increase in pore origin variability in the thickness of cast iron fragments, then the appearance of additional peaks on the approximation curves may indicate the existence of some large cavities or voids. These voids can be aggregated by smaller pores or have a different nature of formation. We assume that such large voids can result from intensive gas extraction from additional admixed components (slags) in the cast iron material [49,53] due to variations in ore crushing and features of ore putting in a melting forge. Note that such large voids are characteristic not only of the fragments from the Selitrennoye settlement but also of the more structurally uniform samples from the Bolgar area.

The analysis of the shape of the separated pores was performed in the approximation of an elongation [47,54,55,56], which represents an ellipsoid shape of pores. The elongation of the pores is the ratio of the semi-axes of the Legendre ellipsoid [57]. A simple description of the average form of pores is the flattening of elongated particles along one of the ellipsoidal axes. The obtained distributions of the elongation parameter of the pores and their average values are presented in Figure 6a. The normal density distribution approximation yields broad asymmetric duplets and triplets (Figure 6b), which may indicate the differences in pore shape and elongation from fragment to fragment. Such a complex form of distribution may indicate several sets of pores inside the cast iron fragments, whose nature is discussed below.

A spatial orientation of the elongated pores can be obtained with an inertia tensor [56,57]. The components of the inertia moment tensor were defined as three orthonormal axes [47,57] for each pore. The *Z*-axis is the rotation axis of the turntable in the tomography setup. We choose the principal axis as the *I*_min_ axes along the *Z*-axis, or the rotation axis in the tomography experiments, and the *I*_max_ axis perpendicular to the *Z*-axis [47].

The obtained stereoplots (Figure 7a–d) are non-uniform, which evidences the presence of the shape fabric [57,58,59]. Such configurations of preferred orientations in the small pores inside cast iron fragments can indicate some alignment of pores along the curved forms on the fragments’ surfaces, which is typical for the shapes of cast iron fragments. It is quite possible that such preferential orientation is due to mechanical effects during forging or hammering, but more likely, this is due to the processes of uneven cooling of cast iron cauldrons.

We mentioned above that the size distributions and elongation parameters indicate at least two groups or types of internal pores in the volume of cast iron products. A similar situation is observed for the orientation of pores in the cast iron fragments. The comparative pole figures for the orientation of small pores (<10 mm) and large ones (>10 mm) are shown in Figure 8a,b, respectively.

Interestingly, different texture fabrics for small and large pores were observed in the cast iron SS-6 fragment. If small pores are characterized by the so-called “belt” texture or fabric, then for large ones, one pole is observed. For the other studied fragments with observed pores, a similar pattern was observed, although the statistical sampling for that pore quantification was poorer. Again, in addition to the size and elongation distributions, the orientation structural features indicated two types of pores, which form in the cast iron fragments. Their difference might indicate a difference in the reasons and mechanisms of their formation [60]. The first group is composed of small pores present in all the studied fragments from the Selitrennoye settlement and the BS-65 sample from the Bolgar settlement. Recently, comparative microscopy and chemical studies of cast iron products [3,4,61] suggested that a historical school of cast iron craft in Bolgar was quite advanced in comparison with the Selitrennoye settlement. It is believed that cast iron products from the Bolgar settlement are of higher quality. Moreover, in the Selitrennoye settlement, there are traces of more oxidation of the cast iron process, which causes increased porosity in the final cast iron products [60,61].

Observed small pores may occur during the casting of medieval products, and their formation can be associated with two processes when cooling a cast iron product. There is a formation of various phases in the Fe-C system [3,4,61,62], including graphite from excess carbon content, and the uneven cooling of the cast of cauldrons. It is known that molds made of a densified sand–clay mixture were used for casting cauldrons [3,4]. These occurred in the solid–liquid state of iron in the temperature range between liquidus and solidus points [4,61]. Therefore, we should expect an uneven spatial distribution within the volume of cast iron fragments. Indeed, a preferential appearance of pores near the outer side of the cauldron fragment was found. Examples of the characteristically reconstructed 3D models for fragments BS-65 and SS-5 are shown in Figure 9a,b, respectively.

More interesting is the fact that both the size and elongation distributions, as well as direct observation, distinguish the presence of large pores or cavities. This is the so-called gas–shrinkage porosity [36,40,61]. When the raw material is milled and loaded into the forge, the small particles of ore matter can get into the fusion matter and can serve as a source of accompanied gas during the melting process. Moreover, if large cavities along with small pores exist in the objects from the Selitrennoye settlement, then only large pores are observed in the cast iron fragments from the Bolgar settlement (Appendix A). Several enlarged slices of the 3D models of the cast iron fragments in the area with large pores are shown in Figure 10. It can be seen that some structural elements with a different neutron attenuation coefficient in comparison with cast iron material were visible next to the large pores. This can be relatively large inclusions from a clay–sand mold for cauldrons.

## 4. Conclusions

Neutron tomography studies of the fragments from cast iron cauldrons of the ancient Golden Horde state were performed. Based on the obtained 3D neutron tomography data, it was possible to identify and separate the internal pores inside these cast iron products. At first sight, the presence or absence of small internal pores can be associated with the location and casting technological processes of the foundries. Analysis of neutron tomography 3D data to obtain the size distributions, some morphological characteristics, and orientation of the internal pores in the cast iron fragments can provide not only qualitative but also quantitative structural markers for cast iron objects. We assume that the versatility of gas shrinkage porosity processes in the cast iron procedures in ancient workshops can provide structural markers to identify the locations of cast iron manufacturers, the presence of additional forging of cast iron products, as well as the features and primary composition of casting molds.

## Figures and Tables

**Figure 1 jimaging-09-00097-f001:**
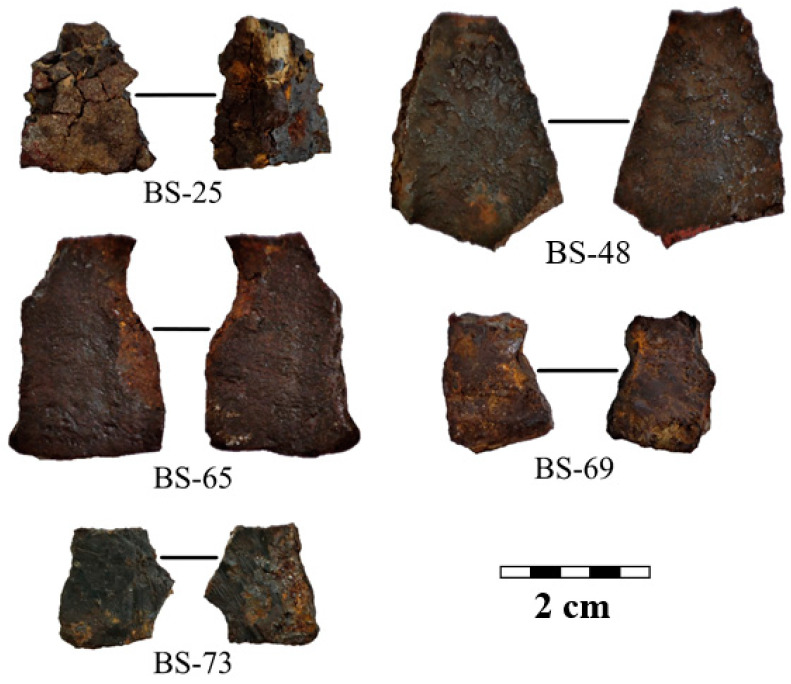
Photos showing the two sides of cast iron fragments of the cauldrons from the Bolgar settlement. A scale bar is presented.

**Figure 2 jimaging-09-00097-f002:**
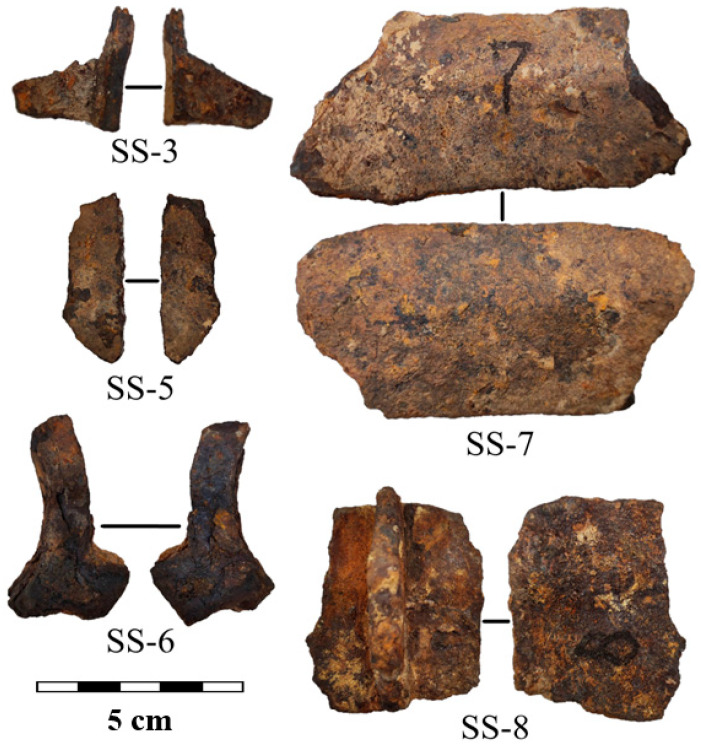
Photos showing the two sides of cast iron fragments from the Selitrennoye settlement. A scale bar is presented.

**Figure 3 jimaging-09-00097-f003:**
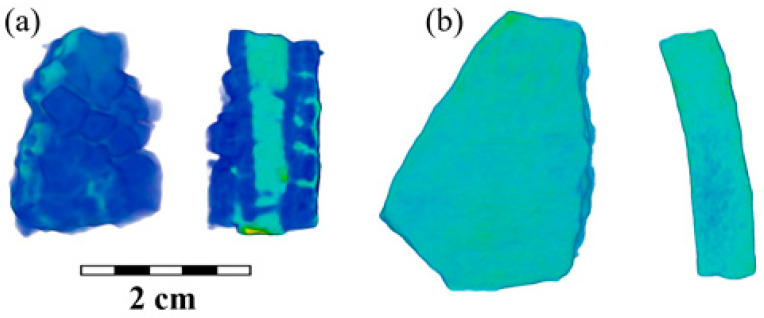
(**a**) The reconstructed 3D model and example of its longitudinal slice for the fragment BS-25. The cast iron regions are labeled in blue, and the possible welding track material is green–blue. (**b**) The reconstructed 3D model and example of its longitudinal slice for the fragment BS-48. A scale bar is presented.

**Figure 4 jimaging-09-00097-f004:**
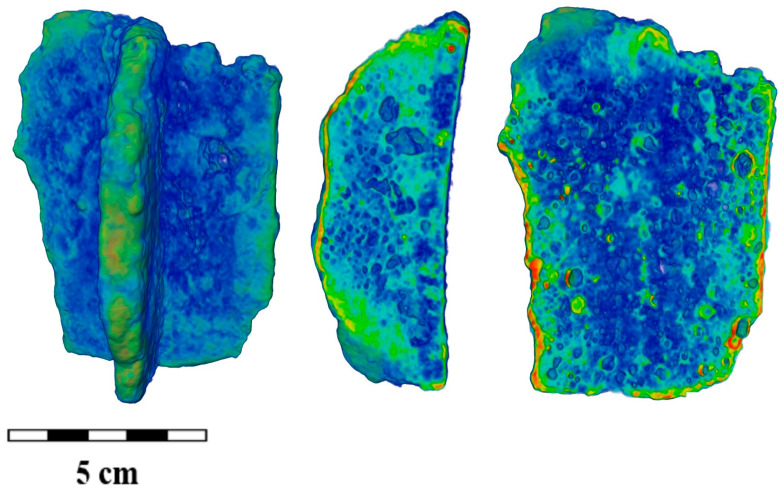
The reconstructed 3D model and some longitudinal and transverse slices of the iron cast fragment SS-8. The cast iron regions are labeled in green–blue color, and the corrosion surface layers are in red. A scale bar is presented.

**Figure 5 jimaging-09-00097-f005:**
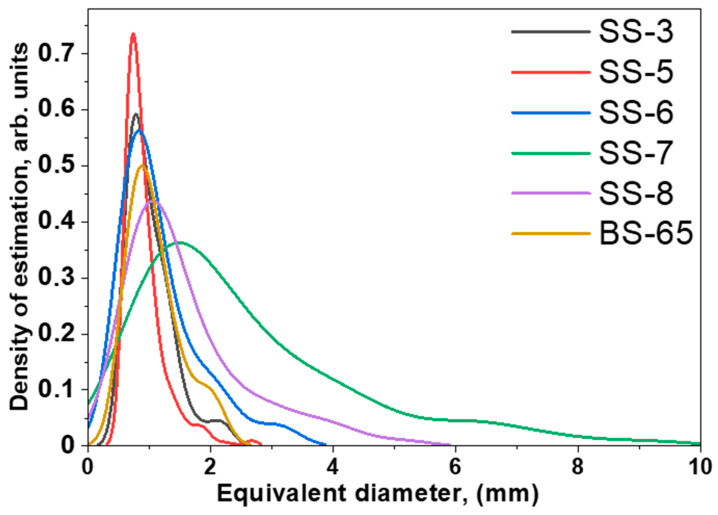
The probability density function for the equivalent diameter distribution of the pores in the selected cast iron fragments.

**Figure 6 jimaging-09-00097-f006:**
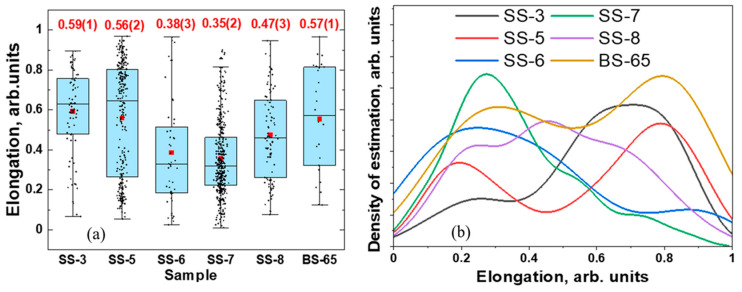
(**a**) A box-view presentation showing the distribution of elongation parameters for selected cast iron fragments. The points are experimental data from the analysis of the 3D models, and the red dots are the mean value of the elongation parameters. These parameters are labeled at the top. The horizontal lines inside the boxes correspond to the median value of the elongation parameter. (**b**) The distribution of elongation of the pores in the cast iron fragments shown as the normal distribution approximation with Kernel Smooth mode.

**Figure 7 jimaging-09-00097-f007:**
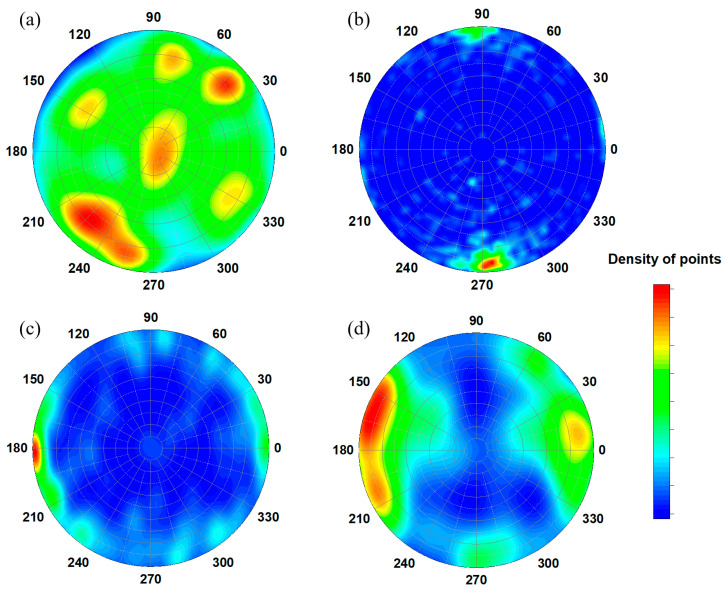
Stereoplots showing the orientation of principal inertia axis *I*_min_ of separated pores inside the iron cast fragments SS-5 (**a**), SS-7 (**b**), BS-65 (**c**), and SS-8 (**d**). The upper hemisphere stereographic projection is shown. The density of point values is coded in the sidebar.

**Figure 8 jimaging-09-00097-f008:**
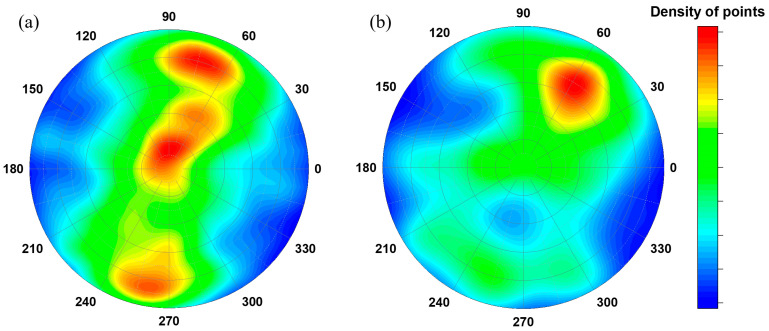
Stereoplots showing the inertia axis *I*_min_ orientation of small (**a**) and large pores (**b**) inside the iron cast fragment SS-6 with respect to the laboratory coordinate system in the tomography experiment. The density of point values is codified in the sidebar.

**Figure 9 jimaging-09-00097-f009:**
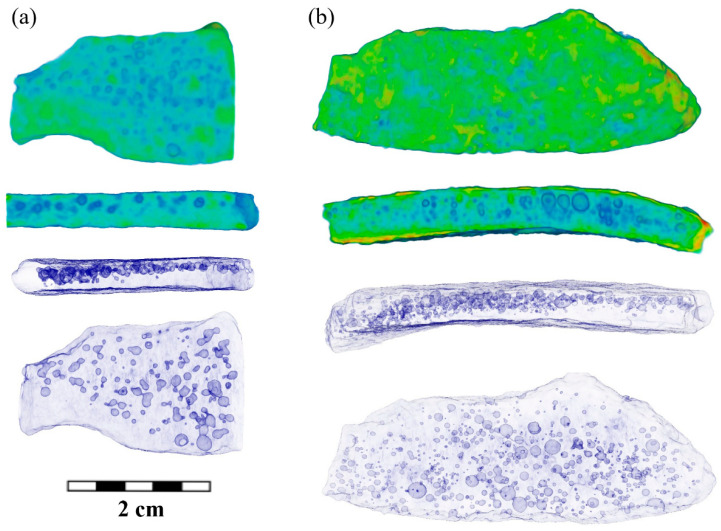
The reconstructed 3D model and some longitudinal slices of the cast iron fragments BS-65 (**a**) and SS-5 (**b**). The cast iron regions are labeled in green–blue color. The pores are presented as sharper images for convenience. The transparent 3D models with pores volumes are presented. The scale bar is shown.

**Figure 10 jimaging-09-00097-f010:**
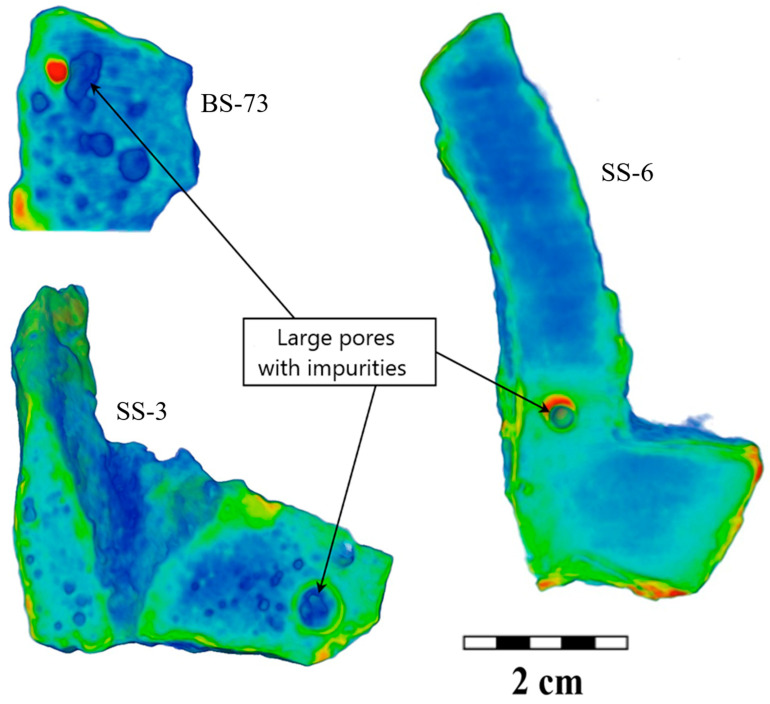
The reconstructed 3D model and several longitudinal slices of the cast iron fragments BS-73, SS-3, and SS-6. The cast iron regions are labeled in green–blue color. The inclusions or contamination impurities are presented in red color. The scale is shown.

**Table 1 jimaging-09-00097-t001:** Structural characteristics of the pores inside the cast iron fragments.

Fragment	Total Volume, mm^3^	Pores Volume, mm^3^	Porosity, %	Mean of Equivalent Diameter, mm	Median of Equivalent Diameter, mm
SS-3	5462.12 (2)	68.2 4(8)	1.25 (1)	1.02 (1)	0.92 (8)
SS-5	5735.67 (5)	420.85 (7)	7.33 (8)	0.92 (1)	0.81 (9)
SS-6	36882.14 (4)	97.03 (9)	0.26 (3)	1.17 (6)	0.96 (9)
SS-7	344044.94 (8)	16961.55 (2)	4.93 (1)	2.69 (3)	2.01 (9)
SS-8	44234.14 (3)	1003.86 (3)	2.26 (9)	1.59 (5)	1.22 (8)
BS-25	1214.76 (8)	0	0.00	-	-
BS-48	3856.79 (5)	0	0.00	-	-
BS-65	21479.66 (8)	65.01 (1)	0.30 (3)	1.09 (2)	1.00 (1)
BS-69	1386.374	1.77 (1)	0.12 (8)	- *	- *
BS-73	1050.08 (6)	13.85 (1)	1.31 (9)	- *	- *

* The number of pores is insufficient for the statistical analysis.

## Data Availability

The data presented in this study are available on request from the corresponding author.

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
