# Peer review of "Structural Features of the Fragments from Cast Iron Cauldrons of the Medieval Golden Horde: Neutron Tomography Data"

_2313-433X, 2023, doi:10.3390/jimaging9050097_

Round 1

Reviewer 1 Report

The authors performed neutron tomography measurements of ten specimens of ancient cast iron cauldrons. Then they obtained the size, elongation, and orientation distributions of internal pores to discuss the results on the viewpoints of a structural marker of the location of cast iron foundries and a feature of the medieval casting process. This reviewer recommends accepting this manuscript for publication after some minor revisions as follows:

Line 79: The authors mentioned, "The average length is in the range of 3 – 20 cm", but the width of the largest sample (SS-7) seems to be about 10 cm. I feel "20 cm" is rather larger than the photograph.

Line 91-92: The correlation between the unit and the definition is inconsistent in the sentence "3.6 x 10^6 n/ (cm2 s) at the neutron wavelength λ = 2.4 Å". The neutron flux is defined in some wavelength range. Therefore, the unit needs to include "/Å", or the wavelength range of integration is indicated.

Line 92-94: The authors used monochromatic neutons by the double cristal monochrometer. The wavelength of the neutrons used in the measurement is quite important infomation in neutron radiography/tomographym measurements.

Figure S2: The same rages of horizontal and vertical axes of the figures would be helpful for comparison. Plotting counts relative to the total counts for the vertical axes might be more helpful.

Line 176: Readers may wonder what parameter was used as a weight to obtain the "weighted distributions".

Figure 6 (a): Readers may wonder why the black dots are distributed left and right sides of the center.

Figure 6 (b): The definition of the vertical axis needs to be clarified. It may be the number of pores relative to the total number, but such values do not larger than 1.

Figure 7: The ranges of scale bars are not the same. More space would be available if one common scale bar were used for all stereoplots by plotting the number of pores relative to the total number.

Figures 7 and 8 look ellipses not circles.

Author Response

Dear Review,

We would like to re-submit our paper

“Structural features of the fragments of cast iron cauldrons of medieval Golden Horde: neutron tomography data”

by

  1. Bakirov, V. Smirnova, S. Kichanov, E. Shaykhutdinova, M. Murashev, D. Kozlenko, A. Sitdikov

Ref. No.:  jimaging-2345174

I would like to sincerely thank Reviewers and Editors for careful reading of the manuscript and providing the useful remarks and comments.

We had made following explanations and corresponded corrections:

 Reviewer 1

The authors performed neutron tomography measurements of ten specimens of ancient cast iron cauldrons. Then they obtained the size, elongation, and orientation distributions of internal pores to discuss the results on the viewpoints of a structural marker of the location of cast iron foundries and a feature of the medieval casting process. This reviewer recommends accepting this manuscript for publication after some minor revisions as follows:

Line 79: The authors mentioned, "The average length is in the range of 3 – 20 cm", but the width of the largest sample (SS-7) seems to be about 10 cm. I feel "20 cm" is rather larger than the photograph.

Thanks for the comment. We corrected to the more truthful value of 3-10 cm.

Line 91-92: The correlation between the unit and the definition is inconsistent in the sentence "3.6 x 10^6 n/ (cm2 s) at the neutron wavelength λ = 2.4 Å". The neutron flux is defined in some wavelength range. Therefore, the unit needs to include "/Å", or the wavelength range of integration is indicated.

You are right, in the experiments we used a monochromator and the neutron flux is given for a certain wavelength value of 2.4 Å. We have reformulated this sentence.

Line 92-94: The authors used monochromatic neutrons by the double cristal monochrometer. The wavelength of the neutrons used in the measurement is quite important infomation in neutron radiography/tomographym measurements.

The neutron wavelength in the neutron imaging experiment was 2.4 Å. This corresponds to our answer above. We have corrected this part of the Experimental Section.

Figure S2: The same rages of horizontal and vertical axes of the figures would be helpful for comparison. Plotting counts relative to the total counts for the vertical axes might be more helpful.

We put the total number of pores on each graph of the Figure S2. The main idea is the shape of the approximation curve by the probability density function. We can represent the distribution not in the number of pores, but in relative frequency, but the shape of the PDF curve and the probability density or density of estimation will not change. It was for comparison that we provide the PDF approximation.

And you are right, but in the Figure S2 we have presented the absolute number of pores for the simple understanding of the unprepared reader, how the calculated curves for Figure 5 were obtained. This Figure shows exactly the normalized distributions, which is why they are presented in the same range. The question of normalized pore quantities is quite debatable. If we change the distributions to the relative number or relative frequency of pores, i.e. normalized by the total number of pores, this will be mathematically correct, but for a wide range of readers the fractional value of the number of pores will not be clear. After all, the context of the publication is not aimed at deep mathematical calculations, but has a more visual character in the search for structural markers for cast iron fragments.

Line 176: Readers may wonder what parameter was used as a weight to obtain the "weighted distributions".

We remove word “weighted”.

Figure 6 (a): Readers may wonder why the black dots are distributed left and right sides of the center.

Horizontal distribution has no physical meaning. This carries only representative functions for a better perception of the density or number of points in the massive array for elongation values. Such a box and points view can be found both in scientific publications (Hoffmann A and et al, Plants. 2023; 12(3):507, https://doi.org/10.3390/plants12030507) and in special discussions (https://priyankadobhal.medium.com/create-a-bar-chart-with-the-individual-data-points-410ad61876fc).

Figure 6 (b): The definition of the vertical axis needs to be clarified. It may be the number of pores relative to the total number, but such values do not larger than 1.

No, the vertical axis in Figure 6b does not correspond to the relative number of pores. It's not even a possibility. This is the PDF -probability distribution density. The most interesting thing is that this value can be higher than 1. However, in accordance with the above discussion about the mathematical apparatus of the PDF, I would like to note that in the context of a publication on the research of archaeological objects, a discussion about probability and density of probability will be superfluous. It should only be noted that, in fact, the absolute values themselves are as important as the shape of these distributions is important. And in our case, we have a probability density distribution that indicates two types of pores inside the object. We have made appropriate changes in the axe caption to the Figure so as not to provoke readers to unreasonable conclusions.

Figure 7: The ranges of scale bars are not the same. More space would be available if one common scale bar were used for all stereoplots by plotting the number of pores relative to the total number.

As discussed earlier, this is a probability density function - PDF, not a probability. It is incorrect to compare the absolute values of this value on stereo plots. We have removed the absolute values from the radial plots of PDF to reduce the discussion. Here again, we need exactly the shape of the orientation pole figures with the formation of a fabric. We have made appropriate changes in the axe caption to the Figure so as not to provoke readers to unreasonable conclusions.

Figures 7 and 8 look ellipses not circles.

We tried to make the stereoplots more rounded. This is a representation defect of image formatting.

Sincerely yours,

Kichanov Sergey,

on behalf of co-authors.

Reviewer 2 Report

The paper presents the neutron tomography analyses of cast iron cauldron fragments of the ancient Golden Horde. The writing is clear and concise, the results are presented adequately.

The literature review is thorough, but there are a lot of old citations.

I am listing a few articles for the authors' kind consideration:

1. It has nothing to do with cast iron object / fragments, but it has a robust description about the evaluation of pores / voids: Gait, J., Bajnok, K., Szilágyi, V. et al. Quantitative 3D orientation analysis of particles and voids to differentiate hand-built pottery forming techniques using X-ray microtomography and neutron tomography. Archaeol Anthropol Sci 14, 223 (2022). https://doi.org/10.1007/s12520-022-01688-y

2. Pores in spearhead from the Bronze Age: Tarbay et al. Non-destructive analysis of a Late Bronze Age hoard from the Velem-Szent Vid hillfort. https://doi.org/10.1016/j.jas.2020.105320.

I might miss the information, but if not, please provide the spatial resolution of the DRAGON neutron tomography setup. It is important to know which is the smallest void that can be identified within a bulky sample. ("the samples from the Bulgar area have almost no internal pores" (line 138))

In my opinion, the manuscript can be accepted after minor revision.

The English of the paper is good, only a few typos could be identified,

e.g. in line 38 "X-ray fluorescens" -> "X-ray fluorescence"

line 197 - "density of points values is codded"

Author Response

Dear Review,

We would like to re-submit our paper

“Structural features of the fragments of cast iron cauldrons of medieval Golden Horde: neutron tomography data”

by

  1. Bakirov, V. Smirnova, S. Kichanov, E. Shaykhutdinova, M. Murashev, D. Kozlenko, A. Sitdikov

Ref. No.:  jimaging-2345174

I would like to sincerely thank Reviewers and Editors for careful reading of the manuscript and providing the useful remarks and comments.

We had made following explanations and corresponded corrections:

The paper presents the neutron tomography analyses of cast iron cauldron fragments of the ancient Golden Horde. The writing is clear and concise, the results are presented adequately.

The literature review is thorough, but there are a lot of old citations.

I am listing a few articles for the authors' kind consideration:

  1. It has nothing to do with cast iron object / fragments, but it has a robust description about the evaluation of pores / voids: Gait, J., Bajnok, K., Szilágyi, V. et al. Quantitative 3D orientation analysis of particles and voids to differentiate hand-built pottery forming techniques using X-ray microtomography and neutron tomography. Archaeol Anthropol Sci 14, 223 (2022). https://doi.org/10.1007/s12520-022-01688-y

Thanks a lot for the advice. We have included this reference in our manuscript.

  1. Pores in spearhead from the Bronze Age: Tarbay et al. Non-destructive analysis of a Late Bronze Age hoard from the Velem-Szent Vid hillfort. https://doi.org/10.1016/j.jas.2020.105320.

Thanks a lot for the advice. We have included this reference in our manuscript.

 I might miss the information, but if not, please provide the spatial resolution of the DRAGON neutron tomography setup. It is important to know which is the smallest void that can be identified within a bulky sample. ("the samples from the Bulgar area have almost no internal pores" (line 138))

Yes, on the advice of you and another reviewer, we have remade the Experimental section about the DRAGON experimental station. We have changed  “the samples from the Bulgar area have almost no internal large pores”

The English of the paper is good, only a few typos could be identified,

e.g. in line 38 "X-ray fluorescens" -> "X-ray fluorescence"

We correct it.

line 197 - "density of points values is codded"

We correct it.

Sincerely yours,

Kichanov Sergey,

on behalf of co-authors.
